# Prediction of Enzyme Specificity using Protein Graph Convolutional Neural Networks

## Abstract

Specific molecular recognition by proteins, for example, protease enzymes, is critical for maintaining the robustness of key life processes. The substrate specificity landscape of a protease enzyme comprises the set of all sequence motifs that are recognized/cut, or just as importantly, not recognized/cut by the enzyme. Current methods for predicting protease specificity landscapes rely on learning sequence patterns in experimentally derived data with a single enzyme, but are not robust to even small mutational changes. A comprehensive evaluation of specificity requires consideration of the three-dimensional structure and energetics of molecular interactions. In this work, we present a protein graph convolutional neural network (PGCN), which uses a physically intuitive, structure-based molecular interaction graph generated using the Rosetta energy function that describes the topology and energetic features, to determine substrate specificity. We use the PGCN to recapitulate and predict the specificity of the NS3/4 protease from the Hepatitic C virus. We compare our PGCN with previously used machine learning models and show that its performance in classification tasks is equivalent or better. Because PGCN is based on physical interactions, it is inherently more interpretable; determination of feature importance reveals key sub-graph patterns responsible for molecular recognition that are biochemically reasonable. The PGCN model also readily lends itself to the design of novel enzymes with tailored specificity against disease targets.

## 1 Introduction

Selective molecular recognition between biomolecules e.g. protein-protein, DNA-protein (Tainer & Cunningham, 1993), and protein-small molecule interactions, is key for maintaining the fidelity of life processes. Multispecificity, i.e. the specific recognition and non-recogntion of multiple targets by biomolecules, is critical for many biological processes, for example the selective recognition and cleavage of host and viral target sites by viral protease enzymes is critical for the life-cycle of many RNA viruses including SARS-CoV-2 (Vizovišek et al., 2018). Prospective prediction of the sequence motifs corresponding to protease enzyme target sites (substrates) is therefore an important goal with broad implications. Elucidating the target specificity of viral protease enzyme can be used for the design of inhibitor anti-viral drug candidates. The ability to accurately and efficiently model the landscape of protease specificity i.e. the set of all substrate sequence motifs that are recognized (and not recognized) by a given enzyme and its variants would also enable the design of proteases with tailores specificities to degrade chosen disease-related targets.

Most current approaches for protease specificity prediction involve detecting and/or learning patterns in known substrate sequences using techniques ranging from logistic regression to deep learning. However, these black-box approaches do not provide any physical/chemical insight into the underlying basis for a particular specificity profile, nor are they robust to changes in the protease enzyme that often arise in the course of evolution. A comprehensive model of protease specificity requires the consideration of the three-dimensional structure of the enzyme and the energetics of interaction between enzyme and various substrates such that substrates that are productively recognized (i.e. cleaved) by the protease are lower in energy than those that are not.

To encode the topology and energetic features, here we develop Protein Convolutional Neural Networks (PGCN). PGCN uses experimentally derived data and a physically-intuitive structure-based molecular interaction energy graph to solve the classification problem for substrate specificity. Protease and substrate residues are considered as nodes, and energies of interactions are obtained from a pairwise decomposition of the energy of the complex calculated using the Rosetta energy function. These energies are assigned as (multiple) node and edge features. We find that PGCN is as good as or better than other previously used machine learning models for protease specificity prediction. However, it is more interpretable and highlights critical sub-graph patterns responsible for observed specificity patterns. As it is based on physical interactions, the PGCN model is capable of both prospective prediction of specificity of chosen protease enzymes and generating novel designed enzymes with tailored specificity again chosen targets.

## 2 RELATED WORK

In this work, we develop a graph-based deep learning technique for protease enzyme specificity prediction. Here we provide a brief review of previously developed predictive methods for protease specificity landscape prediction and applications of graph-based convolutional neural networks on protein-related problems.

### 2.1 PREDICTION OF PROTEASE SPECIFICITY LANDSCAPE

Current methods to discriminate the specificity landscape of one or more types of protease enzymes, could be classified into two categories, machine learning approaches and scoring-matrix-based approaches (Li et al., 2019). Methods use machine learning methods such as logistic regression, random forest, decision tree, support vector machine (SVM) to predict substrate specificity. The most popular tool is SVM among them, e.g. PROSPER (Song et al., 2012), iProt-Sub (Song et al., 2018), CASVM (Wee et al., 2007), Cascleave (Song et al., 2010). Besides, NeuroPred (Southey et al., 2006) and PROSPERous (Song et al., 2017) applied logistic regression to predict specific neuropeptide specificity (Neuropred) and 90 different proteases (PROSPERous). Pripper (Piippo et al., 2010) provided three different classifiers based on SVM, decision tree and random forest. Procleave (Li et al., 2020b) implemented a probabilistic model trained with both sequence and structure feature information. DeepCleave (Li et al., 2020a) is a tool to predict substrate specificity by using convolutional neural network (CNN). None of methods mentioned above use energy-related features, however we note that some energy terms for the interface were considered in (Pethe et al., 2017) and (Pethe et al., 2018).

### 2.2 GRAPH CONVOLUTIONAL NEURAL NETWORK ON PROTEIN-RELATED PROBLEMS

There are several works proposing or implementing a graph-based convolutional neural network model to solve various protein modeling-related problems. BIPSPI (Sanchez-Garcia et al., 2019) made use of both hand-crafted sequence and structure features to predict residue-residue contacts in proteins. Gligorijevic et al. (2020) proposed a novel model that generated node features by using LSTM to learn genetic information and edge adjacency matrix from contact maps to classify different protein functions. Graph convolutional neural networks are also applied to drug discovery, for either node classification or energy score prediction (Sun et al., 2019). Fout et al. (2017) proposed a graph-based model to encode a protein and a drug into each graph, which considered local neighborhood information from each node and learned multiple edge features for edges between neighbor residues and nodes. Zamora-Resendiz & Crivelli (2019) addressed their model to learn sequence- and structure- based information more efficiently than 2D/3D-CNN for protein structure classification. Moreover, Cao & Shen (2019) and Sanyal et al. (2020) aimed at improving the energy functions used for protein model evaluation by using molecular graphs. Unlike previous work, our approach uses per-residue and residue-residue pairwise energies as features for predicting molecular function.

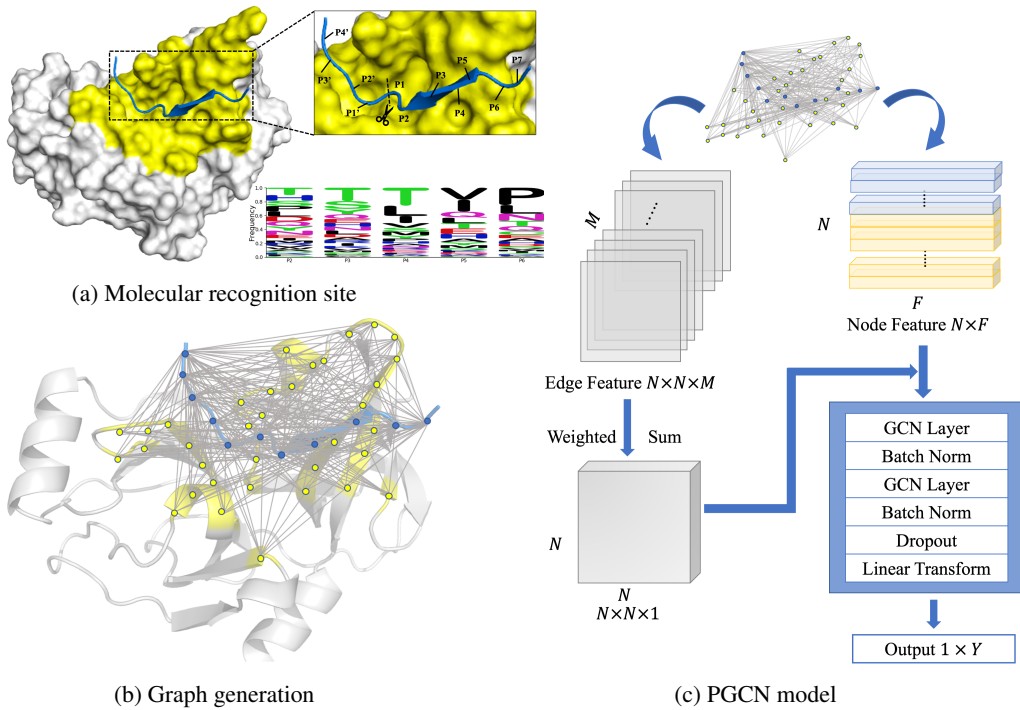

Figure 1: Schematic diagram of a protease with a substrate sample, graph generation and protein graph convolutional network (PGCN).**(a)** Protease - substrate diagram with a sequence logo plot for the specificity landscape of P2-P6 sites. Here, the substrate labeled in blue is 11-amino-acid long, from P7 at N-terminus to P4' at C-terminus and it is cleaved between P1 and P1'. The site labeled in yellow represents all neighbor residues which are 6 angstrom around the substrate. **(b)** Graph generation from the substrate residues (blue dots) and its neighbor residues (yellow dots). Grey lines denote edges. **(c)** PGCN training scheme. In the node matrix, residues are listed in the order of residues starting from substrate residues followed by neighboring residues. In the node matrix, residues are listed in the order of residues from N-terminus to C-terminus, with substrate residues followed by protease residues.

## 3 METHODS

**Overview** We provide protein graph convolutional network (**PGCN**), which models protein-substrate complexes as fully connected graph structures conditioned on terms of both sequences and interaction energies. We generate energy-related features using Rosetta (Leaver-Fay, 2011), a software suite for protein modeling and design, for either single residues or pairwise residues. Substrate is a 7-residue stretch (7-letter string) of amino acids sequences being recognized by a protease active site (Figure 1a). The goal is to predict all motifs that are (are not) efficiently and correctly bound by the enzyme resulting in cleavage (non-cleavage).

### 3.1 PROTEIN GRAPH REPRESENTATION

The protein complex is encoded as a fully connected graph $\mathcal{G} = (\mathcal{V}, \mathcal{E})$, from which substrate and neighboring amino acids make up the nodes, and residue pairwise interactions make up edges (Figure 1b). Each node $v_i \in \mathcal{V}$ contains the one-body features of a single residue, while each edge $e_{ij} \in \mathcal{E}$ contains relational features between a pair of residues. Since residues that are far away from the substrate are less likely to influence specificity and more likely to introduce noise into the model, we only consider a sub-graph $\mathcal{G}'$, including the substrate and neighboring residues to a specified distance around the substrate. The graph $\mathcal{G}'$ consists of two sets: nodes and edges. The features of nodes are encoded as a $N \times F$ matrix, where $N$ is the number of nodes, $F$ is the number of node features. The features of edges are encoded as an $N \times N \times M$ adjacency tensor. (Summary of node and edge features are shown in Table A.4.)

## 3.2 Normalization

Since the stability of the structure depends on the minimum potential energy, we normalize each energy-related feature with the exponential of negative potential to amplify lower energy values and reduce higher energy values, which could be denoted as $Q_i = \exp(-P_i)$, where $P_i$ is $i$th node/edge potential energy.

## 3.3 Multiple Edge Features

Traditionally, the adjacency matrix is used to represent whether a pairs of residues is adjacent or not, which is denoted as $\bar{A} \in \mathbb{R}^{N \times N}$ (Wu et al., 2020) with $\bar{A}_{ij} = 1$ if $e_{ij} \in \mathcal{E}$ and $\bar{A}_{ij} = 0$ if $e_{ij} \notin \mathcal{E}$. This representation is not able to handle multiple edge features, however, and in this work we are incorporating distances and multiple energetic components into the edge features, which Rosetta combines as a weighted sum (Alford et al., 2017). Thus it was necessary for us to flatten the edge feature tensor into a weighted adjacency matrix by summing pseudo total energy for each edge as shown below.

$$E_i^G = \sum_j w_j E_{ij} \tag{1}$$

where $E_i^G$ denotes total energy for an edge, $w_j$ denotes learned weight parameter for the $j$th energy terms and $E_{ij}$ denotes one of the edge energies for the edge. The matrix form of it is shown in Eq.2 later.

## 3.4 Protein Graph Convolutional Network

PGCN learns features of nodes and edges and receives both tuned and learned parameter sets from training data, which is composed of graphs generated from Rosetta-modeled protease-substrate complexes with known class labels (cleaved/uncleaved). PGCN feeds a batch of graphs as input at every step, calculates the loss between estimated and true labels, and updates learning parameters by gradient descent during back propagation.

Here we describe the details of the PGCN architecture. First, PGCN multiplies the adjacency tensor of each graph from training data with a $M$-long learning weight vector $W_A$ to flatten multiple edge features using the weighted sum method. Second, the flattened adjacency matrix $N \times N$, together with the node feature matrix, are fed into a graph convolutional (GC) layer shown as Eq.3. Third, the GC layer is followed by a BatchNorm layer (Ioffe & Szegedy, 2015), which aims at avoiding slow convergence. Next, the output from BatchNorm goes through another GC layer and a BatchNorm layer to continue refining learning parameters. Then, PGCN drops out a proportion of hidden nodes over nodes to avoid the overfitting problem (Srivastava et al., 2014). Unlike normal multi-layer neural networks, PGCN does not count a proportion of hidden nodes out over all nodes using the standard dropout strategy. Instead, PGCN mutes different combinations of hidden nodes for different nodes. Finally, PGCN transforms the output into a $Y$-dimensional vector, which shows the probabilities that the graph is classified into each class over $Y$ classes. Here, the set of tuning hyperparameters is made up of batch size, learning rate, dropout rate and weight decay coefficient. The weight decay coefficient is a part of the L2 regularization term, that multiplies sum of learned weights for the anti-overfitting problem. Learning parameters keep updated through epochs. The trained PGCN model is used for testing, in which test data pass through each layer of the PGCN model but skip the dropout process.

The mathematical expression of PGCN model for one graph is shown below,

$$A' = AW_A \tag{2}$$

$$H_1 = \tilde{D}'^{-\frac{1}{2}} \tilde{A}' \tilde{D}'^{-\frac{1}{2}} X W_X \tag{3}$$

$$H_2 = \tilde{D}'^{-\frac{1}{2}} \tilde{A}' \tilde{D}'^{-\frac{1}{2}} H_1 W_1 \tag{4}$$

where $W_A \in \mathbb{R}^{M \times 1}, W_X \in \mathbb{R}^{F \times C_1}, W_1 \in \mathbb{R}^{C_1 \times C_2}$ are learning weight matrix, $C_1, C_2$ are numbers of hidden nodes for two convolution layers, $\tilde{A}' = \tilde{A}' + I_N$ and $\tilde{D}'$ is a diagonal matrix with $\tilde{D}'_{ii} = \sum_j \tilde{A}'_{ij}$. Eq.2 shows the formulation of adjacency matrix $A'$ from weighted sum of adjacency tensor $A$ over features. In Eq.3 and Eq.4, $\tilde{D}'^{-\frac{1}{2}} \tilde{A}' \tilde{D}'^{-\frac{1}{2}}$ denotes normalization of the adjacency matrix, mentioned in *Normalization* section.

## 4 EXPERIMENTS

We use PGCN to discriminate protease specificity reflected on substrate cleavage, based on two sets of features: a hybrid set which contains both sequence (amino acid types) and energy information, and a set which contains only energy information with categorical features. Cleaving, not cleaving are two main possibilities of protease-substrate interaction. We also consider a ternary classification, wherein a protease-substrate pair exhibited very low cleavage activity in experiments, and can be said to partially cleave a substrate.

We then compare our results with SVM models from (Pethe et al., 2017) and (Pethe et al., 2018), and also compare with currently used machine learning models as mentioned in *Related Work* section. Furthermore, we make analysis on importance of nodes/edges to address PGCN's contribution to potential valuable enzyme design problems.

### 4.1 DATA

We have lists of 5-amino-acid-long substrate sequences (from P2-P6) that were determined in previous experiments (Pethe et al., 2017) to be cleaved, not cleaved, or partially cleaved by the wild type Hepatitis C (HCV) protease, or one of three mutants. HCV mutants had few substitutions, and were homology modeled to have similar backbone geometry. Despite the fact that the mutants have only 1-3 substitutions, they have significantly altered specificity landscapes. The database consists of 66,441 protein-substrate complexes and we trained 46,505 samples into four separate models for three classes, and trained on 27,475 samples of all training samples into another four PGCN models for two classes, see Table 1 for details. The proportion of training and testing are the same among all models, which is 70%: 30%.

Table 1: Number of samples for different data

| # Samples | Two Classes | | Three Classes | |
|---|---|---|---|---|
| | Train | Test | Train | Test |
| WT | 5139 | 2203 | 6106 | 2621 |
| A171T | 9246 | 3962 | 15169 | 6501 |
| D183A | 8304 | 3560 | 12350 | 5294 |
| R170K/A171T/D183A | 4786 | 2052 | 12880 | 5520 |

### 4.2 PROTEIN COMPLEX MODELING

We modeled all P6-P2 substrate sequences in the context of an 11 amino acid (P7-P4') peptide with all non-variable residues consistent with the canonical HCV substrate sequence, bound to the HCV protease in active (cleaving) conformation, based on the crystal structure (PDB:3M5L) (Romano et al., 2010) from RCSB Protein Data Bank (PDB) (http://www.rcsb.org/) (Berman et al., 2000). Producing the models was done with Pyrosetta (Chaudhury et al., 2010), a Python-based interface to Rosetta, and involved changing the side chains of the substrate to match the experimental sequence, then minimizing the complex using FastRelax (Tyka et al., 2011).

### 4.3 PROTEIN GRAPH GENERATION

We considered residues of the core substrate (P2-P6), and neighbor residues within a 10Å shell of the core substrate as nodes of our input graph (34 nodes in total). All graphs are in the format of PyTorch 1.4.0 FloatTensor (Paszke et al., 2019).

PGCN used a more informative and concise feature set than other machine learning methods. PGCN with hybrid feature encoding mode includes all features described in Table A.4, and PGCN with energy-only features is the same set, excluding the one hot encoders of amino acids. All the other methods have hybrid feature encoding mode to include one hot encoders for P6-P2 amino acid types of substrates and energy terms for them are four coarse-grained potential energy terms instead (Pethe et al., 2017).

### 4.4 TRAINING

As we mentioned above, PGCN with 2 GC layers is used to predict substrate specificity landscapes. Each layer has 20 hidden nodes, and non-linear ReLU term (Glorot et al., 2011) follows each graph convolution layer after implementing BatchNorm (Ioffe & Szegedy, 2015). For the training process of PGCN, we use a cross entropy loss function, a Adam optimizer (Kingma & Ba, 2015) and nonzero weight decay and dropout rate. We compared feature encoding of PGCN with that of other five machine learning methods. We used the Scikit-learn 0.20.1 (Pedregosa et al., 2011) to implement logistic regression (lr), random forest (rf), decision tree (dt) and support vector machine (SVM) classification, and Tensorflow 1.13.1 (Abadi et al., 2016) for artificial neural network (ANN). The ANN model in this experiment is one fully connected layer with 1024 hidden nodes and allows a dropout rate between 0.1-0.9.

### 4.5 NODE/EDGE IMPORTANCE

The determinants of protease specificity have not been isolated from the set of many contributing forces; for example, the most stable complexes do not necessarily correlate with substrate recognition. In order to derive biological insights about important residues or relationships between pairs of residues that contribute to discrimination, we perturbed the values of each node/edge term over all test samples and inspected how much the test accuracy drops, enabling us to efficiently determine the relative importance of each node and edge. To be specific, we perturbed values of each node feature $F_{i,k}$ simultaneously across all samples for the importance detection of each node $i$ and perturbed values of each edge $M_{i,j,k}$ simultaneously across all samples for the importance detection of each edge $(i, j)$. The upper and lower triangular of the adjacency matrix should change simultaneously with the same randomness scheme, since it is a symmetric matrix. Therefore, the number of total perturbations should be $N + \frac{N(N-1)}{2} = \frac{N(N+1)}{2}$ if considering all $N$ nodes and edges. The formula for measuring accuracy loss is given by Relative Acc = (Original Acc − Perturbed Acc)/Original Acc.

Table 2: Accuracy table for models based on feature settings for binary classification

| Methods | Wild Type | | A171T | | D183A | | Triple | |
|---|---|---|---|---|---|---|---|---|
| | Hybrid | E-only | Hybrid | E-only | Hybrid | E-only | Hybrid | E-only |
| Logistic Reg | 92.19 | 76.08 | 95.96 | 79.38 | 89.21 | 69.27 | 92.2 | 70.71 |
| Random Forest | 91.68 | 73.62 | 95.85 | 73.25 | 87.7 | 65.63 | 91.05 | 65.63 |
| Decision Tree | 86.79 | 74.63 | 91.54 | 76.58 | 83.88 | 68.29 | 87.77 | 69.54 |
| SVM | 92.87 | 75.85 | 95.84 | 78.85 | **89.44** | 69.66 | 92.79 | 70.57 |
| ANN | **93.19** | 76.49 | **96.44** | 79.86 | 89.55 | 69.75 | **93.08** | 70.61 |
| PGCN | 92.73 | **90.46** | 96.19 | **95.41** | 88.88 | **88.17** | 92.30 | 90.64 |

Table 3: Accuracy table for models based on feature settings for ternary classification

| Methods | WT | | A171T | | D183A | | Triple | |
|---|---|---|---|---|---|---|---|---|
| | Hybrid | E-only | Hybrid | E-only | Hybrid | E-only | Hybrid | E-only |
| Logistic Reg | 79.97 | 63.3 | 80.08 | 52.45 | 68.45 | 47.26 | 71.85 | 65.65 |
| Random Forest | 80.57 | 56.34 | 81.53 | 52.49 | 69.22 | 48.22 | 72.14 | 61.31 |
| Decision Tree | 70.97 | 62.3 | 73.4 | 52.47 | 62.67 | 48 | 62.03 | 63.97 |
| SVM | 82.18 | 63.68 | 83.69 | 53.76 | 71.23 | 46.88 | 69.2 | 59.67 |
| ANN | **84.32** | 64.75 | **84.63** | 53.76 | **71.99** | 47.56 | **73.73** | 65.80 |
| PGCN | 82.93 | **78.50** | 83.59 | **79.71** | 71.08 | **69.04** | 73.71 | 72.34 |

# 5 RESULTS

## 5.1 FEATURE SET GENERATION

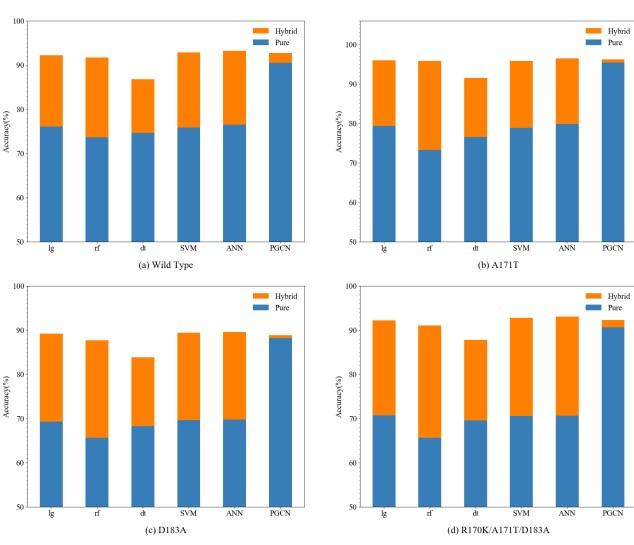

Figure 2: Accuracy of different models based on different features for binary classification. The length of an orange bar denote the closeness of accuracy.

We compare two different feature encoding modes of PGCN (energy-only and energy+sequence "hybrid" features) with that of current five machine learning methods for both two classes and three classes. Accuracy based on the hybrid feature set is not significantly different across different models for each data, except that decision tree reaches at least 9% lower accuracy. ANN always reaches the highest accuracy among all data, for example, up to 96.4% for two classes of HCV with A171T mutation are predicted correctly and up to 84.6% for three classes of it.

When it comes to energy-only encoding, PGCN always performs the best, up to 95.4% for two classes of A171T HCV, about 15% higher than the best accuracy among other machine learning methods, see Table 2[1]. 3. From the Figure 2 and A.5 (in Appendix), it is easy to see that accuracy values of energy-only feature encoding drop heavily down to a similar level for all five machine learning methods, which shows that coarse-grained energies used in other machine learnign models are not informative enough for label classification. PGCN based on energy-only feature encoding almost recovers the accuracy based on hybrid feature encoding.

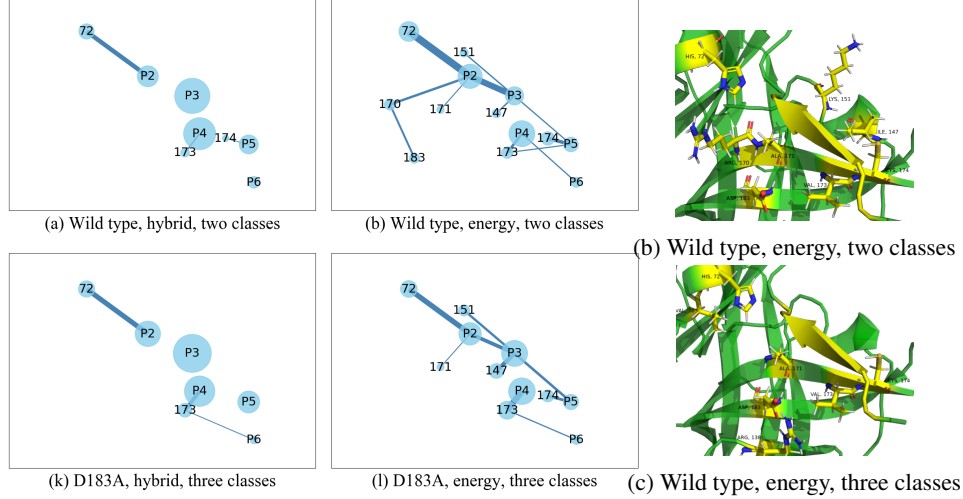

(a) Importance diagram.

Figure 3: **(a)** Parts of the importance diagram for wild type HCV, binary classification and D183A HCV, ternary classification. Sizes of nodes, widths of edges are proportional to importance scores. See The full importance diagram **(b)** Real protein structure examples with highlighted important nodes. Important nodes are labeled in the form of "amino acid type, residue index".

---

[1]"Triple" equivalent with "R170K/A171T/D183A", "Logistic Reg" equivalent with "Logistic regression"

## 5.2 NODE/EDGE IMPORTANCE FOR PROTEASES

When comparing network diagrams for hybrid (sequence+energy) feature encoding with those for energy-only feature encoding, the energy-only PGCN found the largest contributions were from substrate nodes, except that two classes for wild type HCV in and three classes for D183A HCV have two significant edges, P2-72 and P4-173. Among all important edges, edges that connect residues of the substrate with residues of the protein stand out to be the main contribution of importance. First, models detect P2-72 as an important edge, always within top 2 of all edges. Residue 72 is the catalytic residue, (see Figure 3b). Next, a set of important connections is between the substrate and the residues of a beta-sheet of the enzyme, such as P4-173, P5-174, P2-171. We suspect that the importance of these interactions are due to the fact that this beta sheet serves as a template for the substrate and aids in positioning the scissile bond in the active site. In most cases, edges between substrate notes were of lower influence. Moreover, several edges between residues both from the protease show importance as well, such as 170-183 of the wild type binary model, 138-183 of the wild type ternary model, and 171-183 of the A171T ternary model. These residues may not interact directly with the substrate, but as they form the secondary shell around the binding pocket, they likely impact the stability of substrate binding.

## 6 DISCUSSION

In general, PGCN performs impressively well in recapitulating specificity profiles, especially in models using only energy-based features. Following are some avenues for further improvement of our model.

**More divergent input samples** To generalize PGCN beyond the Hepatitis C NS3/4 protease, more extensive and specific set for other proteases, such as TEV protease (Li et al., 2017; Packer et al., 2017) could be useful. These will allow sampling different enzyme-substrate sequence space as well as chemical enviornments, likely leading to a more robust model.

**Imbalanced data** The number of samples for each class are imbalanced and proportions of number of samples in classes vary especially for ternary classification. For example, the proportion of number of samples in cleaved, partially cleaved and uncleaved classes for wild type HCV is 1:1:3, while the proportion for R170K/A171T/D183A is 1:3:1. This may arise a problem because many machine learning models assume balanced data as the input. In this case, the model may underestimate minority class(es) (Mirza et al., 2019). We tried oversampling strategy (Ke et al., 2018) various machine learning models, and found that it somewhat improves accuracies of certain classes. We would like to further explore different class imbalance learning (CIL) strategies to see if they improve PGCN performance.

## 7 CONCLUSION

In this work, we implemented a protein graph convolution network (PGCN) to classify protease-substrate pairs as either yielding substrate cleavage, partial cleavage, or non-cleavage. Using Rosetta, we generated a structural model for each protease-substrate complex, which we converted into a fully connected graph that encoded potential energies for each single residue and each pair of residues. Using the subgraph that includes the bound peptide and neighboring protease residues, we trained the PGCN to predict the behavior of the interaction. We found that the PGCN reaches equivalent accuracy of other machine learning methods using the combination of sequence and energetic features. Furthermore, we demonstrated that variable importance analysis on the PGCN could be used to identify the nodes and edges most influential in determining protease specificity. This method has the potential to enable better prediction and eventually design of engineered proteases with targeted substrate specificity. Codes for this work are available at https://github.com/Nucleus2014/protease-gcnn-pytorch.

### ACKNOWLEDGMENTS

Thanks to Wenfa Lu, Xiao Li for helpful discussions and supports.

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

# A   APPENDIX

Table A.4: Features for nodes and edges

| TYPE | FEATURES | DESCRIPTION |
|---|---|---|
| Node | aa | One hot encoders for amino acid type |
| | fa_atr | Lennard-Jones attractive potential |
| | fa_rep | Lennard-Jones repulsive potential |
| | fa_sol | Lazaridis-Karplus solvation potential |
| | lk_ball_wtd | Asymmetric solvation potential |
| | fa_elec. | Coulombic electrostatic potential |
| | hbond | Hydrogen bonding potential |
| | is_substrate | 1 if the node belongs to the substrate; otherwise, 0 |
| Edge | fa_intra_sol_xover4 | Intra-residue LK solvation energy |
| | fa_intra_rep | Lennard-Jones repulsive energy between pairwise residues |
| | rama_prepro | Ramachandran preferences of backbone angles |
| | omega | Omega dihedral of the backbone |
| | p_aa_pp | Probability of amino acid type at backbone angles |
| | fa_dun | Side-chain conformation potential |
| | ref | Reference potential of pairwise residues |
| | covalent_bond | 1 if pairwise residues form a covalent bond; otherwise, 0 |
| | intramolecular | 1 if one residue from the substrate, the other from the protein; otherwise, 0 |

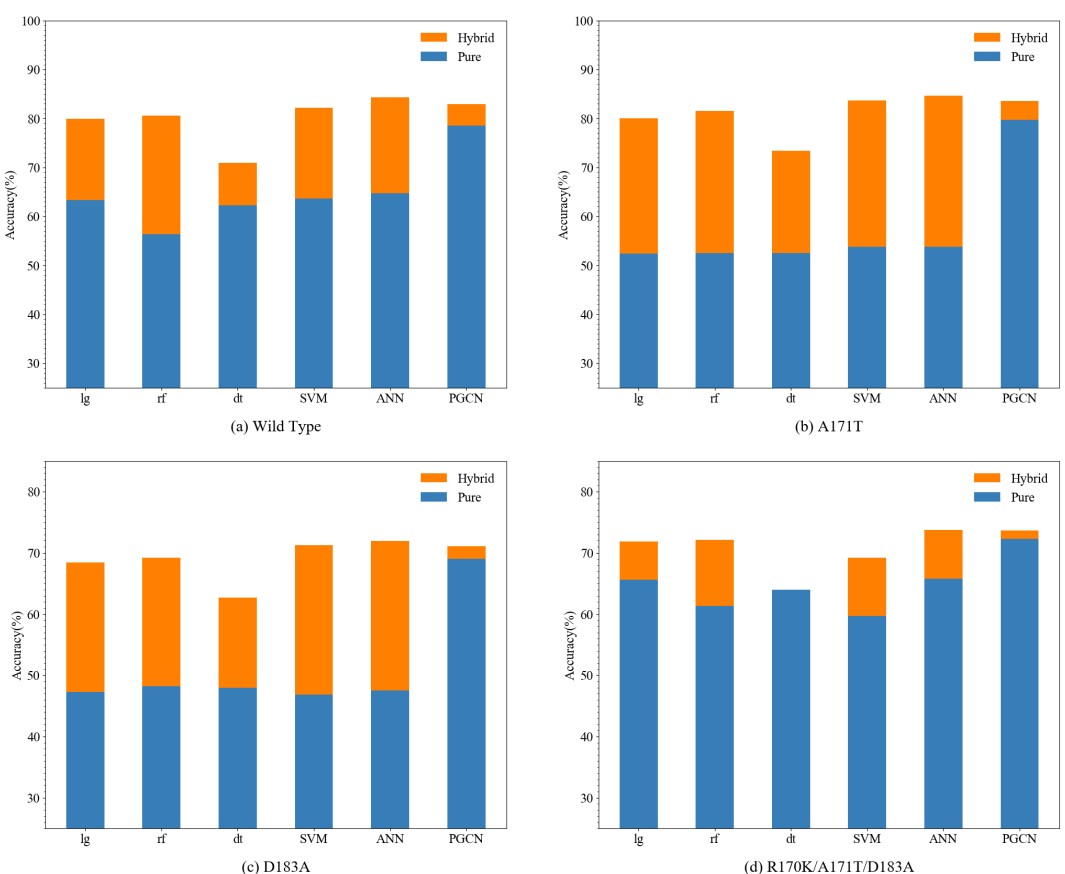

Figure A.4: Accuracy of logistic regression (lg), random forest (rf), decision tree (dt), support vector machine (svm), artificial neural network (ann) and PGCN based on either "hybrid" or energy-only feature encoding for ternary classification. See the caption of Figure 2 for annotations.

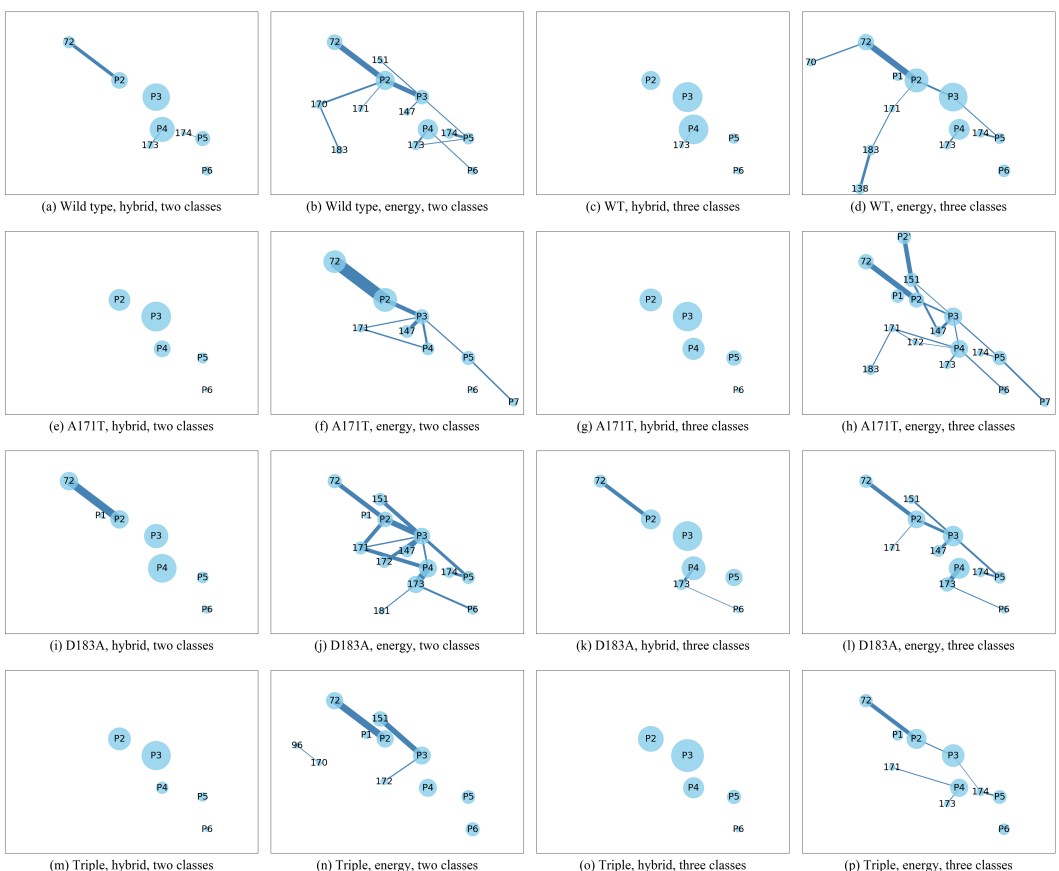

Figure A.5: The whole importance diagram for PGCN on WT, A171T, D183A, R170K/A171T/D183A proteases for either hybrid features or energy-only features.

