# OpenReview forum: "Prediction of Enzyme Specificity using Protein Graph Convolutional Neural Networks"
_ICLR.cc/2021/Conference — Reject_

### Official Review · AnonReviewer1 · 2020-10-26
**Interesting problem but not at the level of ICLR**

**Rating:** 3
**Confidence:** 5

**Review:**

In the work, the authors applied graph convolutional neural networks to predict enzyme specificity. The problem is very important in biology, which even can be used for designing new drugs. Essentially, the authors generated energy-related features using Rosetta and then built the graph using the molecular structure. Then, they applied the existing GCN to the problem. In the evaluation section, they compared with Logistic Regression, Decision Tree, Random Forest, SVM, ANN, to show the superiority of the method.

However, the manuscript has the following major flaws.
1. There is no novelty in the methodology part. They only applied the existing GCN model. The manuscript is not for the audience of ICLR.
2. The baselines are too weak. They only compared with the classic ML methods. They did not compare with the state-of-the-art methods to predict the binding affinity. The authors can find a lot of related literature.
3. Even compared with the classic algorithms, the proposed method did not outperform them by a large margin.
4. They provide the link to the Github repository, which makes the submission not anonymous anymore. That's the reason that I did not provide related works in comment #2.

---

### Official Review · AnonReviewer4 · 2020-10-26
**Official Blind Review #1**

**Rating:** 4
**Confidence:** 4

**Review:**

**Summary**
The paper proposes a new method for calculating the specificity of proteases towards their substrates. The method constructs a graph with node and edge features from the Rosetta molecular energy function, and uses this as the basis for a graph convolutional neural network. The authors report performance on-par on better than existing methods, and highlight interpretability as a potential advantage of the method.

**Strengths**
 o The manuscript addresses and important problem
 o The method claims to produce state-of-the-art performance with a new method.
 o By basing the method on physical features derived from an established energy function, the method can provide some degree of explanation for the predictions made.

**Weaknesses**
 o The paper is primarily an application of existing machine learning methodology, rather than introducing a new method.
 o It is unclear whether the baselines in the paper truly reflect the current state-of-the-art, and a proper train/validation/test split was employed during training.
 o The notation used in the paper is not consistent.

**Recommendation**
I recommend this paper be rejected. In my view, this paper reads more like a machine learning application than a methodological contribution. The presented technique is based on established graph neural network methods, and in the areas were it potentially deviates from this standard it provides insufficient motivation and discussion (see examples below). Since there is no systematic exploration of the methodological contributions, I do not feel that there are sufficient lessons learned in this paper to constitute a valuable contribution to a Machine Learning conference.

**Supporting arguments for recommendation**
Examples with insufficient discussion/motivation of methodological choices:
 o An identify matrix seems to be added to the edge-feature matrix A' (page 4, bottom), but it is unclear why this choice was made - and why it makes sense - since these matrices contain weighted sums of (normalized) energies.
 o A custom dropout technique is introduced, where combinations of hidden nodes are "muted". It is not discussed why this choice was made, and how it compares to standard dropout.
 o Results are reported for two scenarios: A hybrid scenario and a pure energy scenario. However, these choices are not motivated.  Are there cases where we know the energies but not the sequences?  Does Rosetta even allow for calculation of energies without knowing the sequence?

The language used in the paper is also confusing at times. For instance, the term "adjacency matrix" is used to refer to an edge-feature matrix, and they discuss generalization as an "anti-overfitting problem". Also, the edge features that they introduce seem to be referred to by at least three different names, E (eq 1), A (eq 2), and Q/P (top of page 4), and edges are referred to by both single and double indices.

I'm also not confident about how significant the reported results are. While Page 5 mentions a training and test set on Page 5, it is not clear how the hyper-parameters were tuned (no mention of a validation set). Finally, while they do compare their method to several methods from the literature, it was not quite clear to me whether their reimplementations were *identical* to the methods from the literature - or simplified versions.

Finally, it is quite striking that the simple ANN baseline performs so well in the "Hybrid scenario". The authors should at least discuss why this might be.

**Questions to the authors**
Page 5. "We use PGCN... based on two sets"
What is the motivation for studying these two cases. Is the amino acid sequence not always available?

Page 5. "We then compare our results with SVM models from..."
Are these the exact same implementations, or did you merely use *some* SVM implementation. Have you confirmed that your implementations can reproduce the originally reported results of these competing methods?

Page 2. "None of methods mentioned above use energy-related features."
Page 5. "All the other methods have bybrid feature encodings model to include ... and energy terms..."
It seems that these two statements are contradictory. Could you clarify?

Page 5: "The proportion of training and testing..."
Which dataset to you use for optimizing the hyperparameters of the model. Do you have a validation set?

**6. Additional feedback**
Page 1: "non-recogntion" (typo)
Page 2, line 2: "Protein Convolutional Neural Networks (PGCN)". Considering writing "Protein Graph Convolutional Network" here, so that it fits with the abbreviation.
Page 2, "Methods use machine learning methods...". Is there something missing in this sentence?
Page 2, "None of methods mentioned above" -> "None of the methods mentioned above"
Page 2, "using LSTM" -> "using an LSTM"
Figure 1 caption. The last line was a bit confusing to me - since it described the order of the "node matrix" twice. Should one of these have been the "edge matrix"?
Page 3, "Substrate" -> "The substrate"
Page 4, "\tilde A' = \tilde A' + I_N". I assume the first element to the right should be A'?
Page 6. "non linear ReLU term". This is not really a "term", is it?

---

### Official Review · AnonReviewer2 · 2020-10-28
**PGCN performs worse than ANN?**

**Rating:** 4
**Confidence:** 3

**Review:**

This work presents a protein graph convolutional neural network (PGCN) which feeds features from Rosetta through a graph CNN to predict substrate specificity.

Strengths
- Related work is concise but explanatory
- Incorporating features from Rosetta into a graph-based neural network is creative. This is a contribution / idea that could be used in other paper as well.

Weaknesses
- The architecture is described well but there are not any ablations or explanations of how the authors converged on this architecture. Could the results be improved by iterating on the architecture?
- The tested baselines are logistic regression, random forest, decision tree, SVM, and ANN. However, there are no baselines from previous literature, so it is difficult to place this work in the context of the field.
- In Table 2 and Table 3, the default ANN from Tensorflow performs better than the PGCN. In that case, what is the advantage of the PGCN?
- The authors compare a "hybrid" (energy + sequence) to a "pure" approach (energy). The PGCN performs better than other methods in the "pure" setting, but all models perform similarly in the "hybrid" setting. Is there a practical reason to ever use to the "pure" setting? If we are able to get energies in Rosetta, than we likely have the sequence available to us. If the authors can answer this, my next question would be why is PGCN barely affected by incorporating the sequence whereas the other methods see a great improvement?
- From the abstract and introduction, it seems that the authors set up the problem in this way so that they could generalize to new enzymes. However, this is not tested in the paper.

Additional
- In the introduction, the authors should be more clear about "substrate sequence motifs" - do they mean primary or tertiary structure?
- In equation 1, why is a simple weighted sum of edge features used? The individual components could be useful to the network. Why not provide the features directly?

Overall, I was excited to see Rosetta features incorporated into a neural network. However, the evaluation, results, and model development are weak. These would need to be improved if the paper is to be accepted to ICLR.

---

### Official Review · AnonReviewer3 · 2020-10-29
**PREDICTION OF ENZYME SPECIFICITY USING PROTEIN GRAPH CONVOLUTIONAL NEURAL NETWORKS**

**Rating:** 3
**Confidence:** 4

**Review:**

This paper uses a structure-based molecular interaction graph generated from the Rosetta interaction energy function to develop protein graph convolutional neural networks (PGCN) that predict enzyme substrate specificity. 	The authors clearly describe the goal of being able to accurately model the substrate specificity of enzymes such as proteases, including both a description of those substrates that the enzyme recognizes in addition to those substrates that it does not recognize. They propose that this model should capture the energetics of interactions between the enzyme and potential substrates, such that substrates recognized by the enzyme are assigned lower energies by the model than substrates that are not recognized by the enzyme. To address this challenge, they propose a protein graph convolutional neural network, in which the enzyme and substrate residues are modeled as nodes, while interaction energies obtained via a pairwise decomposition of the energy of the enzyme-substrate complex are considered as node and edge features.

The authors provide some survey of recent literature that covers protein-substrate interactions. They can improve by discussing the contributions of papers on this subject from a wider range of research labs - the majority of papers cited in this section feature a common author. Similarly in the related work on graph convolutional networks for protein-related problems the authors should cite work such as e.g. the graph convolutional models for protein ligand interaction prediction from Torng and Altman 2019 among others.

The authors present results for both 'hybrid' and 'energy-only' models. The energy-only feature encoding for the other machine learning models is not clearly described, and it is not clear why the energy-only feature encoding is of interest - the authors do not describe any context in which this encoding would be used in preference to the hybrid sequence and energy feature encoding, which performs better. However, for the energy-only feature encoding the model developed in this paper always performs the best - it is important for the authors to explain why this result is of interest to the reader, since the performance using this encoding is always worse than the performance of multiple other models using the hybrid encoding. For the hybrid encoding and overall, either the ANN or SVM models perform best.

An important question is whether the considerably more detailed energy feature encoding of the PGCN model effectively contains information about the amino acid sequence, making the one-hot sequence encoding that is in present in the hybrid encoding but absent from the energy-only encoding redundant. This would explain why PGCN does comparatively well in the energy-only setting, compared to the much simpler coarse grained energy terms used by the other models. Overall it is very unclear what value the PGCN model and featurization adds.

---

### Decision · Program_Chairs · 2021-01-07
**Final Decision**

**Decision:**

Reject

**Comment:**

All four referees have indicated reject. Severe points of criticism have been raised, concerning the lacking novelty, the  experimental setup and the significance and interpretation of results. I fully agree with the reviewers in all important points, so I recommend rejection.